Complex groundwater flow systems as traveling agent models

López Corona Oliver 1 7 olopez@astro.unam.mx
Padilla Pablo 2
Escolero Oscar 3
González Tomas 4
Morales-Casique Eric 3
Osorio-Olvera Luis 5 6
1 Posgrado en Ciencias de la Tierra, Instituto de Geología, Universidad Nacional Autónoma de México , México D.F. , Mexico
2 IIMAS, Universidad Nacional Autónoma de México , México D.F. , Mexico
3 Instituto de Geología, Universidad Nacional Autónoma de México , México D.F. , Mexico
4 Instituto de Geofísica, Universidad Nacional Autónoma de México , México D.F. , Mexico
5 Posgrado en Ciencias Biológicas, Facultad de Ciencias, Universidad Nacional Autónoma de México , México D.F. , Mexico
6 Departamento de Matemáticas, Universidad Nacional Autónoma de México , México D.F. , Mexico
Minasny Budiman
7 Current affiliation: Theoretical Astrophysics, Instituto de Astronomía, Universidad Nacional Autónoma de México, México D.F., Mexico

Electronic publication date: 2014 Oct 16
Publication date: 2014
Volume: 2
Electronic Location ID: e557
Received 2013 Sep 8; Accepted 2014 Aug 12
Copyright: © 2014 López Corona et al.
Copyright year: 2014
Copyright holder: López Corona et al.
License: This is an open access article distributed under the terms of the Creative Commons Attribution License, which permits unrestricted use, distribution, and reproduction in any medium, provided the original author and source are credited.
License URL: https://creativecommons.org/licenses/by/3.0/

Keywords: Hydrogeology, 1/f noise, Quantum game theory, Complex systems, Spatially extended games

Funding: CONACYT This work was supported by a CONACYT fellowship within the Earth Science Graduate School at UNAM. The funders had no role in study design, data collection and analysis, decision to publish, or preparation of the manuscript.

==============================
Analyzing field data from pumping tests, we show that as with many other natural phenomena, groundwater flow exhibits complex dynamics described by 1/f power spectrum. This result is theoretically studied within an agent perspective. Using a traveling agent model, we prove that this statistical behavior emerges when the medium is complex. Some heuristic reasoning is provided to justify both spatial and dynamic complexity, as the result of the superposition of an infinite number of stochastic processes. Even more, we show that this implies that non-Kolmogorovian probability is needed for its study, and provide a set of new partial differential equations for groundwater flow.

Introduction

Pink or 1/f noise (sometimes also called Flicker noise) is a signal or process with a frequency spectrum such that the power spectral density is inversely proportional to the frequency (Montroll & Shlesinger, 1982; Downey, 2012). This statistical behavior appears in such diverse phenomena as Quantum Mechanics (Bohigas & Schmit, 1984; Faleiro et al., 2006; Haq, Pandey & Bohigas, 1982; Relanyo et al., 2002), Biology (Cavagna et al., 2009; Buhl et al., 2006; Boyer & López-Corona, 2009), Medicine (Goldberger, 2002), Astronomy and many other fields (Press, 1978). Recently the universality of 1/f noise has been related with the manifestation of weak ergodicity breaking (Niemann, Szendro & Kantz, 2013) and with statistical phase transition (López-Corona et al., 2013).

In Geosciences the idea of self-organized criticality (SOC) associated with 1/f power spectrum showed to be important for example in modeling seismicity (Bak, Tang & Weisenfeld, 1987; Bak & Tang, 1989; Bak & Chen, 1991; Sornette & Sornette, 1989). The basic idea of SOC is that large (spatially extended) interactive systems evolve towards a state in which a minor new event can have dramatic consequences. In seismicity this means that earthquakes contribute to organize the lithosphere both in space and time (Sornette, Davy & Sornette, 1990). In this context, the lithosphere may be understood as an unstable and non-linear system made of hierarchy of interacting blocks and in which dynamics has a characteristic 1/f signal (Keilis-Botok, 1990).

A particular active research field in Geoscience is the study of groundwater, which may be considered as a complex dynamic system characterized by non-stationary input (recharge), output (discharge), and response (groundwater levels). For example, groundwater levels in unconfined aquifers never reach steady state and may vary over multiple spatial and temporal scales showing fractal scaling characterized by inverse power law spectra (Zhang & Schilling, 2004; Jianting, Young & Osterberg, 2012). For a review on grounwater transport see Dentz et al. (2012).

Spectral analysis has proven to be a powerful analytical tool for the study of variations in hydrologic processes. Ever since Gelhar (1974) studied temporal variations of groundwater levels for the first time with spectral analysis, it has been widely used. Spectral densities have been used in the study of: the earth tides effect on water level fluctuations (Shih et al., 2000; Maréchal et al., 2002); temporal scaling in discharge (Tessier et al., 1996; Sauquet et al., 2008); scaling in hydraulic head and river base flow (Zhang & Schilling, 2004; Zhang & Li, 2005; Zhang & Li, 2006); water quality variations in space and time domains (Duffy & Gelhar, 1985; Duffy & Al-Hassan, 1988; Kirchner, Feng & Neal, 2000; Schilling et al., 2009). Using the Detrended Fluctuation Analysis (DFA) method, Zhongwei & You-Kuan (2007) have proved that groundwater levels exhibit 1/f behavior for large time scales.

The groundwater flow process may be considered as the motion of agents (water particles) in a heterogeneous medium (Tranouez, Bertelle & Olivier, 2001; Cortis & Knudby, 2006; Park et al., 2008). This problem is analogous to the model of traveling agents presented in Boyer & López-Corona (2009). In that model, the agents are frugivorous animals who feed on randomly located vegetation patches, in a similar way to anomalously diffusing particles in a physical context. The displacement patterns of a variety of animals as albatrosses, bumblebees, primates, gastropods, jackals, seals and sharks, among others (Viswanathan et al., 1999; Ramos-Fernández et al., 2004; Seuront, Duponchel & Chapperon, 2007; Atkinson et al., 2002; Austin, Bowen & McMillan, 2004; Sims et al., 2008) involve many spatio-temporal scales and are sometimes well described by Lévy walks. This is the case of the traveling agents of the model referred to Boyer & López-Corona (2009). A good review on biological aspects of the subject may be found in Miramontes, Boyer & Bartumeus (2012) and for Lévy process see Shlesinger, Klafter & Wong (1982), Klafter, Blumen & Shlesinger (1987) and Lomholt et al. (2008).

The frequent occurrence of pink noise in a seemingly unrelated set of physical systems has prompted an extensive search for common underlying physical principles (Miller, Miller & McWhorter, 1993). In this paper we present a heuristic reasoning for the emergence of 1/f noise in groundwater and propose a new set of groundwater equations for flow in complex media (see Supplemental Information 1).

The traveling agent model

Let us consider a two-dimensional square domain of unit area with N fixed, point-like food patches randomly and uniformly distributed. Each patch contains an amount of food k.

Initially, an agent is located on a patch chosen at random. Then the following deterministic foraging rules are iteratively applied at every time step:

(i) The agent located at patch i feeds on that patch, the fruit content decreasing by one unit: ki → ki−1.

(ii) If ki has reached the value 0, the agent chooses another patch, j, such that kj/dij is maximal over all the allowed patches j ≠ i in the system, where kj is the food content of patch j and dij the Euclidean distance between patches i and j. With this rule, the next visited patch (the “best” patch) has large food content and/or is at a short distance from i. It was assumed that the travel from i to j takes one time unit.

(iii) The agent does not revisit previously visited patches.

This model produces complex trajectories that have been studied in detail in Boyer et al. (2006) and Boyer, Miramontes & Larralde (2009) and discussed in connection with spider monkeys foraging patterns observed in the field (Ramos-Fernández et al., 2004). Most interesting is the fact that when this model is combined with a forest one, the coupled model exhibits self-organized criticality and 1/f power spectrum for biomass time series (Boyer & López-Corona, 2009).

We propose that it is possible to use an equivalent model to study groundwater flow, conceptualizing it as the motion of water particles (agents) in a hydrogeological medium.

Assume the existence of a scale of support w where porous media properties can be measured. This scale of support is kept constant and is small enough such that, at the scale of the flow domain, w can be represented as a point-like quantity. Let us consider a two-dimensional square domain of unit area with N fixed, point-like Hydrogeological Units (HU) randomly and uniformly distributed. Each HU is characterized by its hydraulic flow potential, defined as Ki = Hi/Ri, where Hi and Ri are hydraulic head and hydraulic resistivity at point i, respectively. Thus Ki has units of time.

Initially, an agent (water particle) is located on an HU chosen at random. Then the following deterministic motion rules are iteratively applied at every time step:

(i’) An agent located in an HU stays there for a dimensionless time T proportional to Kmax/(K + a), where Kmax is the maximum hydraulic flow potential in the domain and a is an arbitrary normalization constant such that Kmax ≫ a. For K → 0 then the waiting time is the maximum possible; for K → Kmax then the waiting time is the minimum possible.

(ii’) When an agent has spent T time in the HUi, it chooses another HUj, such that ΔKij/dij is maximal over all the allowed HU (i ≠ j) in the domain, where ΔKij/dij is the hydraulic flow potential difference between HUi and j, and dij is the Euclidean distance between points i and j. With this rule, the next visited HU has the largest hydraulic flow potential gradient. It is assumed that the travel from i to j takes one time unit.

(iii’) For a particular set of initial and boundary conditions, the agent does not revisit previously visited HU.

With this set of rules, both models, biological and groundwater flow, have the same statistical properties despite representing very different systems and then a direct analogy may be considered.

This traveling agent model exhibits some remarkable properties. Let us define the displacement of an agent R(t) = |R(t + t0)−R(t0)| with R(t) is the location of the agent at time t. For analysis, averages were taken over different and independent realizations. If the hydraulic flow potential K follows an inverse power-law distribution P(K) = cK−β, where c is an arbitrary constant and β is a coefficient that represents the medium homogeneity. When β is large (β ≫ 1) the medium is very homogeneous, meaning that all HU have similar values of hydraulic flow potential. On the contrary when β is small (β ∼ 1) the medium is very heterogeneous, meaning that HUs with high hydraulic flow potential are numerous. The intermediate case is when β = 3 and corresponds to a complex medium where HUs with high hydraulic flow potential are present but they are not so numerous.

Lévy Walks and 1/f Dynamics

In recent works (Eliazar & Klafter, 2009a; Eliazar & Klafter, 2009b; Eliazar & Klafter, 2010) proved that Lévy walks and 1/f are the result of systems which superimpose the transmission of many independent stochastic signals.

With this in mind, we proceeded to investigate if the power spectrum of the agent’s motion follows a 1/f dynamic. We found a non-trivial relationship between the homogeneity coefficient β, the motion of the traveling agent and the type of noise observed. These results (summarized in Table 1) are new and differ from previous work since now the motion of the agents is explicitly analyzed.

Table 1 Relation between media homogeneity coefficient β, type of medium, agent motion, and the noise type observed.

Homogeneity	Medium type	Agent motion type	Displacement noise type	
β = 2	Inhomogenous
Disordered	Random confined	White
Uncorrelated	
β = 3	Complex
Transition point	Lévy
Fractal	Pink (1/f)
Transition point	
β = 5	Homogeneous
Ordered	Brownian	Brown
Highly correlated	

Fifty time series for R(t) were generated using the implemented traveling agents model in Boyer & López-Corona (2009) which we propose is analogous to groundwater flow. Three values of β = {2, 3, 5} were considered, corresponding to disordered, complex and ordered media. Then all the 50 power spectra were averaged and fitted by an inverse power law S(f) ∼ f−λ. White noise correspond to a λ ≈ 0, pink to a λ ≈ 1, and brown to a λ ≈ 2.

These results show that the emergence of pink noise for a traveling agent in a heterogeneous medium depends on its degree of heterogeneity. Thus this dynamical behavior may naturally arise from the motion of agents in a complex medium. The agents may be frugivorous monkeys, and the complex medium a rain forest; or the agents may be water particles and the medium an aquifer with a complex geology. Our results suggest that 1/f noise may be a fingerprint of a statistical phase transition from randomness (low correlation associated with white noise), to predictability (high correlation associated to brown noise) an idea suggested to us by A Frank (pers. comm., 2011) and discussed in Fossion et al. (2010).

Study Case

As part of an academic collaboration between German Karlsruhe Institute of Technology (KIT) and Mexico’s National University (UNAM), pumping tests were performed on a set of urban wells in the metropolitan zone of San Luis Potosi City in Mexico (ZMSLP), which hydrogeology is described in Martínez, Escolero & Kralish (2010) and Martinez, Escolero & Wolf (2011).

The metropolitan area is located approximately 400 km northwest of Mexico City. It lies in the San Luis Potosi Valley in the west-centre of the state of the same name at an altitude between 1,850 and 1,900 m above sea level. The area is flanked by the hills of Sierra San Miguelito to the west and Cerro San Pedro to the east; the hills have an altitude of more than 2300 m. The climate is semi-arid with an average rainfall of 356 mm between 1989 and 2006, an average annual temperature of 17.68 °C, and average annual potential evaporation of approximately 2,000 mm. The San Luis Potosi aquifer system underlies much of the surface endorheic basin. It consists of a shallow aquifer and a deep one, separated by a lens of fine material that permits very little interaction. The shallow aquifer is recharged by rainfall in the valley and the Sierra San Miguelito foothills, as well as by leaks from the urban water system. The deep aquifer is recharged in the Sierra San Miguelito and beyond. The 300 km2 of shallow aquifer underlies the urban zone and its periphery. The thickness of the aquifer is within a range estimated at four to 60 m, while the depth of the phreatic level has been reported in general terms as between five and 30 m. The less deep levels are to be found within the urban zone and they deepen towards the east and northeast in the area of peripheral farmland, following the direction of the flow. The deep aquifer covers about 1,980 km2 and underlies the municipalities of San Luis Potosi and Soledad de G. Sanchez, as well as part of Cerro San Pedro, Mexquitic and Zaragoza. The aquifer consists of granular material and fractured volcanic rock, and is confined over most of the flat part of the basin. Usually, wells tapping this aquifer terminate at a depth of 350–450 m and exceptionally at 700 m.

Figure 1 Power spectra for traveling agents with three values of homogeneity.

First column β = 2, the medium is very inhomogeneous (disordered) and the signal is a white noise. Second column β = 3, the medium is complex and the signal is a pink noise. Third column 5, the medium is very homogeneous (ordered) and the signal is a brown noise. Power Spectrum is taken as Sf≡R˜fR˜−f, where R˜f is the Fourier transformation of the displacement calculated by a Fast Fourier Transformation technique.

Figure 2 Power spectra for three pumping tests in the aquifer of San Luis Potosi City in Mexico.

Drawdown data was acquired in 3 s intervals basis, with a total of 1800 measurements. There are two statistical regimes 101 s to 103 s with 1/f noise statistical behavior, and the second one with periods of seconds or less and a white noise type of signal.

The time series from three pumping well tests, in the shallow aquifer, were analyzed. A pumping test is conducted to evaluate an aquifer by “stimulating” the aquifer through constant pumping, and observing the aquifer’s response (drawdown) in observation wells. The power spectrum from all tests shows that there are two statistical regimes (Figs. 1 and 2). The first regime is characterized by time periods from 101 s to 103 s and 1/f noise statistical behavior, and the second one with periods of seconds or less and a white noise type of signal.

Discussion and Conclusions

Major sources of uncertainty have been identified in groundwater modeling. Model parameters are uncertain because they are usually measured at a few locations which are not enough to fully characterize the high degree of spatial variability at all length scales; thus, it is impossible to find a unique set of parameters to represent reality correctly. Stresses and boundary conditions are also uncertain; the extraction of water through wells and vertical recharge due to rain are not known exactly and they must be provided to the model; lateral boundaries are often virtual boundaries and water exchange through them is usually uncertain. Even model structure can be uncertain because a mathematical model is an approximation of reality and thus some physical processes are not completely known or partially represented (Neuman, 2003). In fact, the problem of characterizing subsurface heterogeneity has been one of the biggest obstacles in constructing realistic models of groundwater flow (Fleckenstein, Niswonger & Fogg, 2006). Koltermann & Gorelick (1996) and De Marsily et al. (1998) present a good review on the subject.

Prediction with classical deterministic process models is constrained by several mathematical limitations. For one side, there is measurement error, non-linearity and sensitivity to boundary conditions (chaos) and on the other side we most face model error and inaccessible or uncertain parameters and variables (Little & Bloomfield, 2010). For these reasons, systematic oversimplifications in groundwater problems have been commonly made, under the assumption that if the most important processes are identified, groundwater flow may be sufficiently characterized.

On the other hand, Kirchner, Feng & Neal (2000) found that long-term, time series measurements of chloride, a natural passive tracer, in runoff in catchments exhibits a 1/f dynamic and later (Scher, Cortis & Berkowitz, 2002) gave a physical model to explain these founding in terms of CTRW.

Significant deviations from standard solutions have been observed in pumping tests (Raghavan, 2004). Moreover, it has been reported that 1/f dynamics are observed in time series of pumping test (Zhongwei & You-Kuan, 2007) and we showed evidence that support their findings. One approach to deal with this anomalous behavior has been to formulate the groundwater flow problem in the continuous time random walk (CTRW) framework (Cortis & Knudby, 2006). Alternatively we propose a traveling agent model for groundwater flow. The model proposed is an analogy of a previous one presented by Boyer & López-Corona (2009) which was used to construct time series for agent’s mean-displacement. In agreement with field results, the model generates 1/f dynamics when the ambient where the agent’s move is complex. For this type of medium, the step length follows a power law distribution P(l) ∼ l−a with a ≈ 2; the waiting time distribution follows a power law y(t) ∼ t−d with d = 2 and the mean displacement a power law (R2) ∼ Tg with g ≈ 1.2 (Boyer & López-Corona, 2009; López-Corona, 2007). If the process was a CTRW then the following relationship should hold g = 2 + d−a and a value of g = 2 would be expected (Klafter, Zumofen & Shlesinger, 1995). This suggests that groundwater flow is even more complex than a CTRW, which, in fact, also occurs in spider monkeys’ foraging process for which g = 1.7 (Ramos-Fernández et al., 2004). In this sense, the model proposed could be a forward step in the study of groundwater flow complexity.

Another advantage of the traveling agent model for explaining the emergence of 1/f is that we may identify in which type of hydrogeological medium this kind of dynamic behavior is observed. We proved that pink noise is present when the environment heterogeneities in which the agents are moving are distributed as a power law with a scaling exponent of β = 3, corresponding to a complex medium. Labat et al. (2011) has pointed out that the complex characteristics of karst aquifers make their exploitation more complicated than other porous or fractured aquifers. These types of aquifers are spatially complex (as our β = 3 medium) groundwater systems characterized by an inherent temporal non-stationarity and nonlinearity of their hydrological response.

Eliazar & Klafter (2009a) and Eliazar & Klafter (2009b) have proven that the 1/f statistical dynamic is originated by the superposition of an infinite number of stochastic processes. This suggests that for complex media (as karstic or rock fractured aquifer) no groundwater modeling simplification is valid. This ambient induces 1/f noise and an infinite number of stochastic processes are in play. Therefore, the assumption that groundwater flow may be sufficiently characterized if the most important processes are identified is no longer valid.

Even more, the results may be interpreted also from a physical standpoint; the observable macroscopic behavior of a hydrogeological system at a given location is the result of the superposition of different physical processes at different scales, such as: diurnal barometric variations that affect groundwater levels, temporal fluctuations in recharge rates, moon’s gravitational effects over the aquifer, tide variations in coastal aquifers, variations in the income flow from rivers and discharge through base flow, temporal increase on total stress due to trains, the effect of extraordinary recharge events provoked by an hurricane presence, and the regime of operation of wells in the area. In Labat et al. (2011) it has been proved, using DFA analysis, that in karstic stream flow fluctuations there are three distinct temporal scale ranges: from 1 h to around 100 h, from around 100 h up to 1 year and scales larger to 1 year. Fluctuations in flow show a clearly anti-correlated behavior on time scales above 1 year, with a slope around 0.3 corresponding to white noise. In the intermediate regime from a few days up to 1 year, a positive Hurst effect is observed, with a slope around 0.8 (almost a 1/f noise) as expected. On time scales below the crossover at a few days, the scaling behavior is highly non-stationary and corresponds to a random walk with positively correlated steps (with a slope around 1.75, near a Brown noise type). The authors explain these findings from a hydrogeological point of view. The first temporal scale, 1 to 100 h, is interpreted as the rapid response of the aquifer (associated with the main drain in the karstic system) to the rainfall; the second temporal scale, 100 h to 1 year, is the global response of the aquifer to rainfall input including the temporal structure of the peak flow; the third temporal scale, larger than 1 year, corresponds to the annual response of rainfall input including the regulation of the discharge via annex systems in the saturated zone. It has also been suggested that an explanation for the scale invariance of groundwater levels involve the system response to constantly changing driving inputs and boundary conditions, including boundaries imposed by management regimes, (Little & Bloomfield, 2010). In this way, the 1/f power spectrum observed in groundwater time series may be originated by both complexity of the geological medium and the presence of complex external factors (as time dependent boundary conditions).

Given this, either we accept that these types of complex groundwater systems are not suited to being modeled or we learn to deal with this infinite superposition of stochastic processes. Once groundwater flow is modeled on a traveling agent framework, we propose to describe it as a spatially extended game. Using this approach we have been able to deduce a set of partial differential equations starting from the discrete description of the model (the details of the derivation are presented as Supplemental Information 1). The probability of finding an agent (water particle) in the position (x, y) at the time t obeys (1) ∂tPx,y,t=divex,y,t∇P,

where e(x, y, t) is the strategy (micro-physics of the flow process) that the agent in (x, y) plays at time t. The strategy in turn obeys the equation (2) ∂tex,y,t=−divD1x,y,tex,y,t+∇2D2x,y,tex,y,t.

While in continuous time random walk approaches few parameters suffice to describe a complex system, Eqs. (1) and (2) introduce field (x, y) dependent diffusion and drift coefficients, and thus represent a quite complex approach. Godec & Metzler (2013) has provided an exact expression for the diffusion coefficient in anomalous diffusion process modeled by Lévy walks under linear response regime.

If you take the simple case when e(x, y, t) is a constant (assuming that the porous medium is relatively constant in the observation time scale, and it is sufficiently homogeneous and isotropic, all of which are common assumptions in hydrogeology), then you recover the classical groundwater flow equation Ss∂h/∂t = k∇2h. Our equations then satisfy the correspondence principle since they recover classical formulation and establish the ground for new insights of groundwater flow process, other porous media transport phenomena and even in Game Theory.

Typically, a system is considered complex when it is constituted from a large number of subsystems that interact strongly enough, but there is another source of complexity that has been widely ignored. A system is also complex when the system itself changes over time in the same scale of its dynamics, which is the case in some karstic aquifers. This second source of complexity is taken into account directly in our equations making a contribution in this respect and might have some important interpretation in Game Theory.

Finally, most interesting, using the traveling agent model described in the method section, we proposed (López-Corona et al., 2013) that 1/f noise is a fingerprint of a statistical phase transition, from randomness (disorder—white noise) to predictability (order–brown noise). In this context, one may interpret Labat et al.’s (2011) results as follows: first temporal scale (from 1 to 100 h) represents a rapid response of the aquifer and should be dominated by random processes (white noise); the second (100 h to 1 year) is the global response of the aquifer to rainfall input including the temporal structure of the peak flow one may be interpreted as a complex (with multiple spatio and temporal scales included) process (1/f noise); and as the third correspond to mean (1 year or more) response is a more predictable process (brown noise). We have then a transition from randomness to predictability consistent with power spectra exponent values. In this way, the results of Labat et al. (2011) is only one example of a universal statistical kind of phase transition.

Supplemental Information

Supplemental Information 1 Electronic Supplementary Materials for Complex groundwater flow systems as a traveling agent models

Click here for additional data file.

Additional Information and Declarations

Competing Interests

Author Contributions

The authors declares there are no competing interests.

Oliver López Corona conceived and designed the experiments, performed the experiments, analyzed the data, contributed reagents/materials/analysis tools, wrote the paper, prepared figures and/or tables, reviewed drafts of the paper, revision of the manuscript.

Pablo Padilla conceived and designed the experiments, analyzed the data, contributed reagents/materials/analysis tools, wrote the paper, prepared figures and/or tables, reviewed drafts of the paper, revision of the manuscript.

Oscar Escolero conceived and designed the experiments, analyzed the data, contributed reagents/materials/analysis tools, wrote the paper, prepared figures and/or tables, reviewed drafts of the paper.

Tomas González analyzed the data, contributed reagents/materials/analysis tools, wrote the paper, prepared figures and/or tables, reviewed drafts of the paper, revision of the manuscript.

Eric Morales-Casique conceived and designed the experiments, analyzed the data, contributed reagents/materials/analysis tools, wrote the paper, reviewed drafts of the paper.

Luis Osorio-Olvera analyzed the data, contributed reagents/materials/analysis tools, wrote the paper, prepared figures and/or tables, reviewed drafts of the paper.

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
