# Peer review of "Complex groundwater flow systems as traveling agent models"

_PeerJ, doi:10.7717/peerj.557_

## Round 0.1 · original submission · Major Revisions

Three reviewers have looked at the paper. The 2nd and 3rd reviewers expertise in physics recommended minor revision, however the 1st reviewer, expert in modelling of transport in porous medium recommended a rejection. I can see the reviewer's point that the authors have not justified how the agent model is appropriate for groundwater flow. The examples (Fig 2) do not show how the model can be applied and used. Although the simulation (Fig. 1) showed similar statistical properties, it doesn't mean the model is applicable directly. You need to fully justify the model and show how it was applied to the pump test data.

Reviewer 1 ·

Basic reporting

No comments

Experimental design

No comments

Validity of the findings

The authors propose an agent based model as an explanation of observed complex dynamics of groundwater pumping tests. Other authors such as Cortis and Knudby (2006) have proposed a continuous time random walk model for transient flow in heterogeneous media, which I believe to be plausible.

The authors' model, however, is a traveling agent model, previous used for monkeys foraging for fruit.

In my opinion, the authors have not been able to give an acceptable explanation of why the assumptions of agent behavior in this model could be relevant or appropriate to model water flow in porous media.

Additional comments

No comments

·

Basic reporting

The authors present and discuss a travelling agent-based model for the modelling of complex dynamic features in tracer diffusion in groundwater. It is a well known fact that the dynamics of chemical tracers in aquifers cannot be described by conventional diffusion-advection schemes. In this work the authors pay particular attention to the widely observed 1/f noise, that is also a typical characteristic for groundwater observations. I find the manuscript generally well written, timely, and of high relevance to the field. After consideration of the points listed below I expect the work to warrant publication in PeerJ.

Experimental design

No comments

Validity of the findings

No comments

Additional comments

(1) While the language used in the text contains some jargon of the groundwater community which I am not fully familiar with, I stumbled across the somewhat sudden turn in the Discussion section. The body of the manuscript considers 1/f noise and Levy walk interpretations, then suddenly quantum game theory appears. If the authors do not insist on this part, I would recommend removing it from the manuscript. At least to me, this is rather confusing. Moreover, while in continuous time random walk approaches few parameters suffice to describe a complex system, Eqs (1) and (2) introduce field (x,y) dependent diffusion
and drift coefficients, and thus represent a quite complex approach.

(2) Scher et al (2002) is mentioned in the text but does not appear in the References.

(3) Recent literature on 1/f noise should be included: Godec and Metzler, Phys Rev E 88, 012116 (2013) with an exact expression for the diffusion coefficient in the zero frequency limit of the power spectrum; Niemann et al, Phys Rev Lett 110, 140603 (2013) with some general remarks on 1/f noise. Moreover, the Levy walk search contribution by Lomholt et al, Proc Natl Acad Sci USA 105, 11055 (2008) should be mentioned. Finally, the following review on grounwater transport should be cited: Dentz et al, Adv Wat Res 49, 13 (2012). Levy walks were originally introduced in Shlesinger et al, J Stat Phys 27, 499 (1982), see also Klafter et al, Phys Rev A 35, 3081 (1987). Please mention.

Reviewer 3 ·

Basic reporting

See below

Experimental design

See below

Validity of the findings

See below

Additional comments

This is an interesting and timely paper, whose publication I shall endorse once an adequate revision addressing the following points is resubmitted:

1. Describe what a “forest model” is, and provide a reference (line 87).

2. Explain the motivation for the hydrological model: while I was immediately convinced that the foraging model is a reasonable “toy model”, I was not as convinced in the case of the hydrological model. Also, why should condition (iii’) hold?

3. Why is the inverse power-law potential K assumed? Why the exponent 3 is the “intermediate case”?

4. The conclusions conveyed in Table 1 require solid explanations. How
do the authors deduce these conclusions based on only three exponent values? Is the exponent 3 indeed a transition point? Maybe the pink noise holds for some sub-range of exponents? Is the classification actually to the exponent ranges (0,3), {3}, and (3,infinity)? If so, this has to be convincingly established.

5. The Eliazar-Klafter references on 1/f noises should be reinforced by: I. Eliazar and J. Klafter, Universal generation of 1/f noises, Physical Review E 82 (2010) 021109.

6. Can the authors link their “pink noise” to the following paper:
E.W. Montroll and M.F. Shlesinger, On 1/f noise and other distributions with long tails, PNAS 79 (1982) 3380-3383.‏

7. The fact that subscripts are not used (for example $ki$ instead of $k_i$) is rather disturbing. Please fix this throughout the paper.

8. Please proof-check the paper, as I spotted several typos (e.g. lines 116 and 147).

---

## Round 0.2 · accepted · Accept

The authors have revised the paper accordingly. I would suggest that the authors make the review open.